# Molecular Characterization of a Long-Term Survivor Double Metastatic Non-Small Cell Lung Cancer and Pancreatic Ductal Adenocarcinoma Treated with Gefitinib in Combination with Gemcitabine Plus Nab-Paclitaxel and mFOLFOX6 as First and Second Line Therapy

**DOI:** 10.3390/cancers11060749

**Published:** 2019-05-29

**Authors:** Oronzo Brunetti, Giuseppe Badalamenti, Simona De Summa, Angela Calabrese, Antonella Argentiero, Livia Fucci, Vito Longo, Domenico Galetta, Pia Maria Soccorsa Perrotti, Rosamaria Pinto, Daniela Petriella, Katia Danza, Stefania Tommasi, Francesco Leonetti, Nicola Silvestris

**Affiliations:** 1Medical Oncology Unit, IRCCS Istituto Tumori “Giovanni Paolo II”, Viale Orazio Flacco, 65, 70124 Bari, Italy; argentieroantonella@gmail.com; 2Department of Surgical, Oncological and Oral Sciences, Section of Medical Oncology, University of Palermo, 90127 Palermo, Italy; giuseppe.badalamenti@unipa.it; 3Pharmacogenetics and Molecular Diagnostic Unit, IRCCS IstitutoTumori “Giovanni Paolo II”, Viale Orazio Flacco, 65, 70124 Bari, Italy; desumma.simona@gmail.com (S.D.S.); rosamaria.pinto@hotmail.it (R.P.); danielapetriella@libero.it (D.P.); danzakatia@gmail.com (K.D.); s.tommasi@oncologico.bari.it (S.T.); 4Department of Radiology, IRCCS IstitutoTumori “Giovanni Paolo II”, VialeOrazio Flacco, 65, 70124 Bari, Italy; acalabrese22@gmail.com (A.C.); pia.perrotti@libero.it (P.M.S.P.); 5Histopathological Unit, IRCCS IstitutoTumori “Giovanni Paolo II”, VialeOrazio Flacco, 65, 70124 Bari, Italy; livia.fucci@gmail.com; 6Medical ThoracicOncology Unit, IRCCS IstitutoTumori “Giovanni Paolo II”, VialeOrazio Flacco, 65, 70124 Bari, Italy; vito.longo79@tiscali.it (V.L.); galetta@oncologico.bari.it (D.G.); 7Dipartimento di Farmacia-Scienze del Farmaco, Universityof Bari, 70125 Bari, Italy; francesco.leonetti@uniba.it; 8ScientificDirectorate, IRCCS IstitutoTumori “Giovanni Paolo II”, Viale OrazioFlacco, 65, 70124 Bari, Italy; n.silvestris@oncologico.bari.it

**Keywords:** pancreatic ductal adenocarcinoma, non-small cell lung cancer, double primary cancers, B7-H3, gefitinib, chemotherapy

## Abstract

The management of multiple primary cancers, an event not so infrequent in oncology practice, is a critical issue due to the lack of literature. In this study, we reported the case of a patient with non-small cell metastatic lung cancer (NSCLC) and pancreatic ductal adenocarcinoma (PDAC) who received gefitinib in combination with gemcitabine plus nab-paclitaxel and with mFOLFOX6 in first and second line, respectively. It achieved a progression-free survival and a28-months overall survival (OS) for NSCLC and PFS-1 and OS of 20 and 13 months, respectively for PDAC. Moreover, the combination of gefitinib and chemotherapy treatmentsshowed a good safety profile. Given the insignificant frequency of this case, we performed a molecular characterization of both neoplasms with the aim to investigate the existence of particular activated pathways and/or similar immunological mutations. It is interesting to note that two neoplasms shared a common mutation ofthe B7-H3 gene, with the consecutive impairment of its expressed protein. In both PDAC and NSCLC, the expression of this protein was associated with a worse survival rate. Since B7-H3 is an anti-apoptotic protein, the reduction of its expression or function should justify a pro-apoptotic activity with a leading justification of the long survival of the patient considered in this report.

## 1. Introduction

The Surveillance, Epidemiology, and End Results (SEER) Program indicates that multiple primary malignant tumors includea percentage ranging from 1% for an initial liver primary diagnosis to 16% for the initial primary bladder carcinoma [1]. In particular, pancreatic ductal adenocarcinoma (PDAC) associated with double primary tumors shows an incidence variable from 0.75% to 20.0% [2,3,4,5]. The management of these clinical situations is a critical issue due to the lack of data with the exception of single cases or small series [6,7,8]. So far, a patient-oriented approach would be advisable with the choice of activecancer therapies with the worst or active prognosis for both cancers.

The use of anti-Epidermal Growth Factor Receptor (EGFR) Tyrosine Kinase anti-Inhibitor (TKI) plus chemotherapy is still controversial both in non-small cell lung cancer (NSCLC) and in PDAC mutated from EGFR [9]. In particular, several studies failed to demonstrate an advantage in the survival rate of this combination in advanced NSCLC not previously treated [10]. Recently, the combination of gefitinib plus carboplatin/pemetrexed has been shown to be effective in patients with advanced mutation-positive EGFR at a newly diagnosed stage III/IV and a recurrent NSCLC [11]. Moreover, in a phase 1b study the addition of erlotinib to gemcitabine plus nab-paclitaxel in patients with advanced PDAC demonstrated a good efficacy but showed that it was not tolerated at the standard dosage of a single-agent [12].

Herein, we report a case of patient with a metastatic NSCLC in treatment with gefitinib, who received a diagnosis of metachronous PDAC treated simultaneously to anti-EGFRwith gemcitabine plus nab-paclitaxel and with mFOLFOX6 in first and second line, respectively. In spite of the two metachronous aggressive cancers, this patient is achieving long term survival due to the combination of anti EGFR and chemotherapy.The insignificant infrequency of this case prompted us to perform a molecular characterization of both neoplasms with the aim to investigate the existence of particular activated pathways and/or similar immunological mutations that would justify the good response to the selected therapies.

## 2. Experimental Section

### 2.1. Patient Record

Treatment courses and clinical features of this patient were collected at the National Cancer Institute “Giovanni Paolo II” of Bari, Italy (n° 19 of 18 March 2016). The patient signed an informed consent for the study which was approved by the local Ethical Committee.

### 2.2. DNA Extraction

DNA was isolated from 6 µm thick formalin-fixed, paraffin-embedded sections after deparaffinization in xylene, using the QIAamp DNA FFPE Tissue Kit (Qiagen, Hilden, Germany). A custom panel was designed using an Ion Ampliseq Designer tool (www.ampliseq.com). The primer panel covered 95.27 kb and included the entire coding regions of 41 genes involved in several immune check-points, inflammation, and B and T cells activation. The panel contained two primer poolsincluding 261 and 253 amplicons, respectively, with size ranging from125 to 275 bp. Libraries were generated using 10 ng of DNA sample and the amplicons were barcoded and amplified using the Ion Ampliseq™ Library kit 2.0 (Thermo fisher, Waltham, MA, USA) and Ion Xpress™ barcode adapters kit (Thermo Fisher, Thermo fisher, Waltham, MA, USA) according to the manufacturer’s instructions. In addition, the CE-IVD Oncomine solid tumor DNA kit was used in order to detect KRAS mutations. It consists of a single pool of primers and associated reagents to perform multiplex PCR for preparation of amplicon libraries for sequencing hotspots and targeted regions in 22 genes. The quantification of libraries was made by real-time PCR using the Ion Library Quantitation Kit (Thermo Fisher). The Ion PGM Template OT2-Hi-Q View 200 Kit (Thermo fisher, Waltham, MA, USA) was used to perform samplePCR emulsion and enrichment of the IonSphere, according to the manufacturer’s instructions. The Ion PGM Hi-Q View Sequencing Kit was used for sequencing reaction (Thermo Fisher) and the Ion 318 Chip Kit v2 (Thermo Fisher) was used for sequencing on the Ion Torrent PGM sequencer. All generated reads were aligned to the human genome hg19 using the Torrent Suite Server.2.3 (Thermofisher, Waltham, MA, USA).

### 2.3. Variant Calling

The data coming from the custom panel were processed initially using the Ion Torrent platform-specific pipeline softwareTorrent Suite v5.4.0.46, Thermofisher, Waltham, MA, USA) to generate sequence reads, trim adapter sequences, and filter and remove poor signal-profile reads. Initial variant calling from the Ion AmpliSeq sequencing data was generated using Torrent Suite Software v5.4.0 with the plug-in “variant caller 5.4.0.46”. In order to eliminate errors in base calling, the Somatic-High Stringency parameters setting was used in order to generate the final variant calling. Data from the NGS custom panel were also analyzed by a custom pipeline to verify their reliability. In detail, aligned BAM files of tumor samples and pool of healthy controls were downloaded from Torrent Server and analyzed through Vardict algorithms [13]. Then variants were filtered through VCFfilter from vcflib library. In detail, somatic variants were called when matching the following conditions: DP > 50, VD > 20 and QUAL > 30. Germline variants were called when DP > 50, VD > 20, QUAL > 30 and AF > 0.4. Bcf tools [14] were used to exclude germline variants using a pool of healthy controls (bcf tool sisec). Then, the callset was intersected with bcftool to include only variants also called from VariantCaller plugin of TVC. The selected variants were functionally annotated by Annovar version 2016 Feb 01. The LJB* database was used to obtain predictions on deleteriousness from different prediction methods [15]. The CE-IVD Oncomine solid tumor DNA panel data were analyzed through an optimized Ion Reporter workflow, considering only reliable variants with an allelic frequency ≥ 5%. In the variant list, each mutation was verified in the Integrative genome viewer (IGV) from the Broad Institute. Only mutations reported in the COSMIC (Sanger Institute Catalogue of Somatic Mutations in Cancer) database [16] were taken into account and silent or intronic mutations were not reported. String database [17] has been used to build up network revealing relationships among mutated genes.

## 3. Results

### 3.1. Patient’s History

A 64-year-old Caucasian woman was admitted to our Institute with cardiac tamponade. She was a former smoker of2.5 packs/year with a clinical history characterized by a Hashimoto’s thyroiditis and panic attacks disease.

After pericardiocentesis (which resulted positive for cancer cells), the computed tomography scan (CTS) of the chest displayed a nodular opacity (maximum diameter of 1.5 cm with spiculated margins) at the anterior segment of the right upper lobe, suggestive ofa primarylung cancer. An associated parenchymal densification area in the loco-regional area (maximum axial diameter of 2 cm) and other areas of parenchymal densification (diameter ranged from 0.5 to 1 cm) were also present. Two nodular opacities of about 0.3 cm in diameter were evident in the apical segment of the left lower lobe. Moreover, some lymph nodes were described in the hilar-mediastinalarea (maximum short diameter of 0.5 cm) (Figure 1A).

In January 2017, the patient underwent supraclavicular lymphadenectomy with a diagnosis of advanced poorly differentiated NSCLC-A (cT3N3M1a—stage IV) (Figure 2A). Subsequently, gefitinib (250 mg daily) was started according to the presences of exon 21 (L858R), substitution mutations of EGFR, with a prolonged radiological stable disease (SD) of both lung lesions and mediastinal and supraclavicular lymph nodes.

Eight months later, a CTS of the thorax and abdomen showed duodenal stenosis related to an infiltration by a head-pancreatic mass (Figure 1B). Serum level of CA19.9 was 551 ng/mL. In October 2017, a laparotomy was performed with a gastro-entero anastomosis and multiple biopsies of the visceral peritoneum. Histological examination revealed a moderately differentiated PDAC (cT3N1M1—stage IV) (Figure 2B). Radiological examination confirmed the involvement of the lungs and diaphragmatic lymph nodes in the absence of new parenchymal lesions. The first-line chemotherapy regimen with gemcitabine plus nab-paclitaxel according to von Hoff’s trial [18] with continuation of gefitinib was started. Dose adjustments of both chemotherapeutic drugs were performed according to products’ manufacturing data due to G2 neutropenia toxicity after 4months from the beginning. No dose adjustments were required for gefitinib. This combination therapy resulted to be effective with 13 months of median progression-free survival (PFS)-1 for pancreatic lesion and a partial response as the best radiological response associated with both decreased of Ca 19.9 serum levels (56 ng/mL versus 551 ng/mL) and stability of lung and lymph node lesions.

On November 2018, the onset of hyperbilirubinemia related to the progression of pancreatic lesion required the placement of a biliary endoprosthesis (Figure 1C). At that point, she started second-line chemotherapy with mFOLFOX-6 [19]. Gefitinib administration was continued for the SD of NSCLC (Figure 1D). Again, this combination of systemic chemotherapy with the anti-EGFR TKI resulted well tolerated and effectivewith a radiological stability of pancreatic lesion and lymph node metastases after 20 months. Currently, the patient presents a stable situation for both tumors and she is continuing the combination treatment with PFS of 28 months for NSCLC and an OS and PFS-1 of 20 and 13 months, respectively for PDAC.

### 3.2. Molecular Results

For routinely therapeutic purpose, NSCLC has been analyzed through next generation sequencing (NGS) CE-IVD Oncomine solid tumor DNA panel identifying EGFR:p.Leu858Arg (allelic frequency = 0.19) as an actionable alteration. Nevertheless, PDAC lesion similarly was subjected to the same analysis which detected TP53:p.Gly266Ter and KRAS:p.Gly12Val mutations with allelic frequencies of 0.035 and 0.004, respectively. To gain insights into the immunological mutation status, PDAC and NSCLC samples were both analyzed with a custom targeted panel including 41 genes involved in the major immunological checkpoints and inflammation signaling. Mutations, which survived filtering steps (allelic frequency > 0.05 and deleteriousness predicted by at least 9/16 in silico tools), are listed in Table 1. In particular, we were able to identify a single predicted deleterious alteration in CD276 gene. String networks were built up taking into account the somatic mutational pattern of NSCLC and PDAC samples (Figure 3). Interestingly, the common CD276 alteration is connected to the other somatic mutations, suggesting an interplay between them and, in particular, with CD276, also known as B7H3, immune checkpoint pathway which is involved in immune response and apoptosis.

## 4. Discussion

Although pancreatic cancer with double primary tumors seems to be a rare occurrence, the data reports that this event raises an incidence variable from 0.75% to 20.0% [2,3,4,5]. In particular, in a study evaluating the incidence of locations of PDAC-associated double primary tumors, Shin et al. discovered that most common hystologies were found in the stomach (21.5%), colorectum (20.7%), and lung (14.9%) [2]. Interestingly, in this study, PDAC plus NSCLC patients had a tendency for a longer survival than those with pancreatic cancer only (median, 27.4 months vs. 17.0 months; *p* = 0.092). Moreover, Tagawa et al. [4] reported a case collection of 7NSCLC in patients with a history of PDAC estimating five-year survival rate of 68.6%, which was better than the survival rate of patients with PDAC only. 

These data are in line with the clinical outcome of our patient. It must be noted that she presents a better clinical outcome with PFS and OS of 28 months for NSCLC and PFS-1and OS of 20 and 13 months, respectively for PDAC. These data appear clinically relevant if we consider the advanced stage of both malignancies.

Recently, a phase 1b trial [12] explored the combination of gemcitabine, nab-paclitaxel, and erlotinib. This last drug is a reversible TKI with a similar structure to gefitinib. The authors observed a poor tolerability of this schedule of therapy, with occurrence of grade 3 and 4 neutropenia, thrombocytopenia, dehydration, hypotension, and transaminase elevations. Nevertheless, 50% and 36% of patients display PR and SD, respectively, with mPFS and mOS of 5.4 months and 9.3 months, respectively.

To the best of our knowledge, this is the first time that the combination of these three drugs has been explored.

Our patient did not have grade 3 or 4 side effects, but achieved a longer survival and an ECOG performance status of 0. 

Despite the weak tolerability reported in the phase 1b study, we observed a good safety related in both first and second line settings. Perhaps, combining an anti-EGFR with standard chemotherapy could enhance the anticancer effects, in particular in tumors with a biological activation of RAS pathway. In a phase III study, gemcitabine/erlotinib association demonstrated a positive survival benefit compared to gemcitabinealone [20], even if this combination was not approved in several guidelines for first line PDAC due to the excessive cost of erlotinib and the modest increase of survival.

This intriguing report reports the history of a patient with a double primarywith long survival and a good safety profile obtained with the combination of gefitinib and chemotherapy. So far, a molecular characterization of both neoplasms has been performed with the aim to better explore their mutational status, not only assessing the molecular status for therapeutic purpose but also analyzing a custom gene panel involved in inflammation and immune checkpoint. The aim was to identify alterations which could be able to indicate a biological mechanism for the long survival of our patient.

Interestingly, our results showed that both neoplasms shared an alteration in B7-H3 gene. The detected alteration, predicted in silico as deleterious, regards the Leucine 237 which is replaced by an histidine. This mutation is within a conserved domain, as reported by Uniprot, which is type 1 of type C2 of type Ig, and we could hypothesizethat the substitution of a hydrophobic amino acid with an electrically positive onecouldimpair the functionality of theproteinB7-H3. In the last few years, thisgene has been investigated revealing a key role in immune escape in cancer. Moreover, B7-H3 is able to induce anti-apoptotic and proliferative effects [21].

A recent meta-analysis considering NSCLC patients demonstrated that B7-H3 resulted a marker of tumor aggressiveness and lymph node metastasis. Moreover, upregulation of soluble B7-H3 malignant pleural effusion of NSCLC had been revealed a potential diagnostic biomarker correlated with NSCLC staging [22]. More interesting results have been published for PDAC. Recently, the role as negative prognostic marker of B7-H3 in PDAC has been demonstrated and its inhibition has been tested in cell lines showing its decreased expression and modulation of TLR4 [23]. Inamura et al. evaluated the immunohistochemistry B7-H3 expression in 150 PDAC patients. Their data showed that patients with higher B7-H3 expression had a disease-free survival (DFS) higher than those with lower expression (*p* = 0.0026), concluding that B7-H3 expression may be a useful prognostic biomarker for identifying aggressive, early-stage pancreatic cancer [24]. Li et al. [25] demonstrated that B7-H3 reduces apoptosis related to chemotherapy by delivering signals to pancreatic cancer cells, thereby improving chemoresistance. B7-H3 expression was higher in PDAC tissues than in normal tissues of PDAC. B7-H3 activated by 4H7 reduced gemcitabine-induced apoptosis in PDAC Patu8988 cells. Activation of B7-H3 by 4H7 induced variations in p-ERK1/2, EGFR, and IκB protein levels. In particular, when B7-H3 was upregulated, the expression level of IκB significantly decreased (*p* < 0.05), especially in the gemcitabine-treated group [25]. Moreover, the silencing of the B7-H3 gene improved sensitivity to paclitaxel in in vivo mice breast models [26,27]. These data could confirm that the presence of B7-H3 mutation in both cancers (which impairs the function of this protein) should be the reason behindthe observed good response to chemotherapy addicted to anti-EGFR with a prolonged OS.

## 5. Conclusions

Herein, we analyze and document the rare, very intriguing case of a patient with peculiar characteristics: (1) the long term survival of metachronous double metastatic primary PDAC and NSCLC treated with gefitinib in combination with gemcitabine plus nab-paclitaxel and mFOLFOX6 in first and second line, respectively; (2) the good toxicity profile of combinations of an anti-EGFR with standard chemotherapy that could enhance the anticancer effects, in particular in cancers with a biological activation of RAS pathway; (3) the presence of the mutation of the anti-apoptotic gene B7-H3 which should reduce the production of B7-H3 activity with a potential addicted anticancer-effect of an anti-EGFR plus chemotherapy. Clinical trials assessing these combinations should evaluate potential markers of response or toxicity, in both PDAC and NSCLC patients as well as in double metastatic primary cancers.

## Figures and Tables

**Figure 1 cancers-11-00749-f001:**
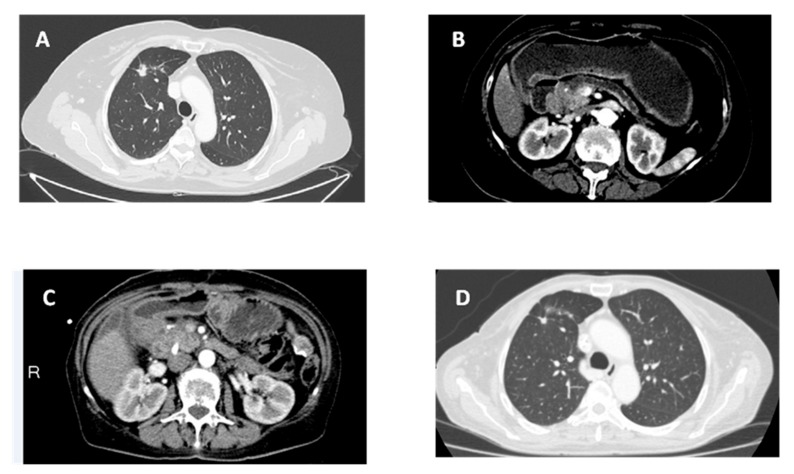
Radiological evaluation of patient with computerized tomography.(**A**) Radiological presentation of primarylung cancer with pericardial effusion. (**B**) Radiological presentation of primary pancreatic cancer with bowel obstruction. (**C**) Radiological presentation of pancreatic cancer at progression. (**D**) Radiological presentation of primarylung cancer in stable disease.

**Figure 2 cancers-11-00749-f002:**
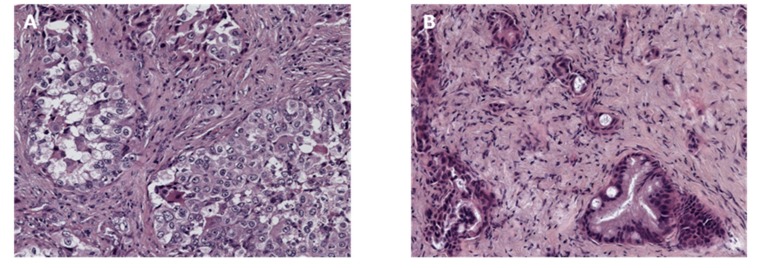
Histological presentation of the two primary tumors. (**A**) Lung adenocarcinoma in haematoxylin and eosin (20×) staining: solid growth aggregates of poorly differentiated cells with large cytoplasm and bulky nucleus with irregular chromatin and nucleolus. (**B**) Pancreatic ductal adenocarcinoma in haematoxylin and eosin (20×) staining: irregular glandular mucinous aggregates (both large and small), immersed in a fibrous stroma, with basal-nucleus polarized cells and sporadic interposed goblet cells.

**Figure 3 cancers-11-00749-f003:**
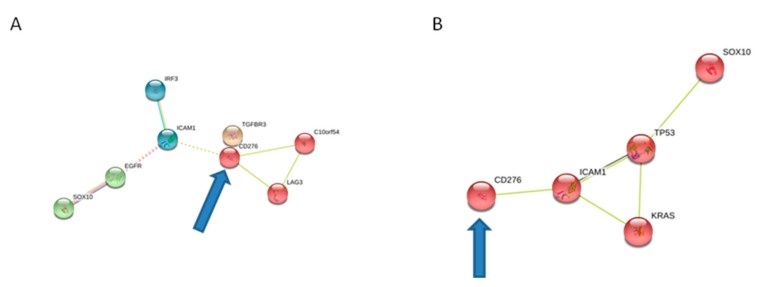
String network including mutated genes in (**A**) lung and (**B**) pancreatic lesion. In the two networks it could be noticed that genes in both of the two pathological settings are connected to CD276, whose mutation is shared by the two tumors.

**Table 1 cancers-11-00749-t001:** Alterationsdetected through custom targeted panel innon-small cell metastatic lung cancer (NSCLC) and pancreatic ductal adenocarcinoma (PDAC) samples.

NSCLC Sample	
TGFBR3 c.* 1C > T	3′UTR
VSIR c.C770T:p.A257V	Non synonimous
LAG3 c.C1360T:p.L454F	Non synonimous
ICAM1 c.G457A:p.G153R	Non synonimous
IRF3 c.G77A:p.W26X	Stop gain
SOX10 c.C1358T:p.T453I	Non synonimous
**PDAC Sample**	
ICAM1 c.C628T:p.Q210X	Non synonimous
SOX10 c.G955A:p.G319R	Stop gain
**Common Mutation**	
CD276 c.T410A:p.L137H	Non synonimous

Mutations that overcome the filtering steps (allele frequency and predicted deleteriousness, as reported in Experimental Section) and their annotation.

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
