# Peer review of "Molecular Characterization of a Long-Term Survivor Double Metastatic Non-Small Cell Lung Cancer and Pancreatic Ductal Adenocarcinoma Treated with Gefitinib in Combination with Gemcitabine Plus Nab-Paclitaxel and mFOLFOX6 as First and Second Line Therapy"

_cancers, 2019, doi:10.3390/cancers11060749_

Round 1

Reviewer 1 Report

This case report presents a patient with a stage IV metastatic NSCLC who subsequently was diagnosed with a second primary cancer namely a PDAC and was treated with gefitinib-gemcitabine and paclitaxel-mFOLFOX6 as first and second lines, respectively, with good response. Molecular profiling of both cancers was performed to explain in part the observed response.

Although the manuscript is well documented and presented there are some points that need attention.

1) A better rephrasing might be useful to clarify that this is the case of a patient with a stage IV metastatic NSCLC who subsequently was diagnosed with a second primary PDAC (metachronous cancer).

2) Staging of the PDAC should be reported.

3) The authors report that the patient underwent a gastro-enteroanastomosis apparently for upper GI obstruction (line 142) and on October 2017 adhesiolysis (line 145) when biopsies were also taken. It is not clear how many times the patient was operated. Is it a single operation? Why biopsies were not taken at the first operation? Is there a second operation and was the reason for that.

4) Attention should be given to linguistic and grammatical errors such as keeping the spaces and using the correct terminology: ilo-mediastinal (line 123), primitive (Figure 1 and line 217, instead of primary), Istological (Figure 2), occlusion (line 142 instead of obstruction), viscerolysis (line 145, do you mean adhesiolysis?).

Author Response

This case report presents a patient with a stage IV metastatic NSCLC who subsequently was diagnosed with a second primary cancer namely a PDAC and was treated with gefitinib-gemcitabine and paclitaxel-mFOLFOX6 as first and second lines, respectively, with good response. Molecular profiling of both cancers was performed to explain in part the observed response.

Although the manuscript is well documented and presented there are some points that need attention.

1) A better rephrasing might be useful to clarify that this is the case of a patient with a stage IV metastatic NSCLC who subsequently was diagnosed with a second primary PDAC (metachronous cancer).

The Authors clarified this concept in the introduction.

2) Staging of the PDAC should be reported.

            The Author reported PDAC stage in the results section line 161.

3) The authors report that the patient underwent a gastro-enteroanastomosis apparently for upper GI obstruction (line 142) and on October 2017 adhesiolysis (line 145) when biopsies were also taken. It is not clear how many times the patient was operated. Is it a single operation? Why biopsies were not taken at the first operation? Is there a second operation and was the reason for that.

            We are very sorry for the misunderstanding. The patient underwent surgery only a time. The Authors clarify clinical events.

4) Attention should be given to linguistic and grammatical errors such as keeping the spaces and using the correct terminology: ilo-mediastinal (line 123), primitive (Figure 1 and line 217, instead of primary), Istological (Figure 2), occlusion (line 142 instead of obstruction), viscerolysis (line 145, do you mean adhesiolysis?).

            The Author revised all the linguistic and grammatical errors with the help of a native English speaker.

Reviewer 2 Report

Brunetti et al has presented a case study of a patient suffering from two primary cancers NSCLC and pancreatic ductal adenocarcinoma, wherein the patient responded well to a combination therapy designed by them. Use of molecular tools to characterize tumor sample was a good step taken to determine the mutations and/or altered pathways. B7-H3 gene mutation, as a common mutation in both primary tumors was an interesting find and would help in designing therapeutic modalities and biomarker development of therapeutic response. 

While the study is interesting for patient care and designing novel therapeutic modalities, especially combination therapies which is getting popular and in most cases, effective, the authors need to pay attention to the severe errors in text editing throughout the manuscript. It makes it difficult to read the manuscript. Please correct these errors. 

Author Response

Brunetti et al has presented a case study of a patient suffering from two primary cancers NSCLC and pancreatic ductal adenocarcinoma, wherein the patient responded well to a combination therapy designed by them. Use of molecular tools to characterize tumor sample was a good step taken to determine the mutations and/or altered pathways. B7-H3 gene mutation, as a common mutation in both primary tumors was an interesting find and would help in designing therapeutic modalities and biomarker development of therapeutic response.

While the study is interesting for patient care and designing novel therapeutic modalities, especially combination therapies which is getting popular and in most cases, effective, the authors need to pay attention to the severe errors in text editing throughout the manuscript. It makes it difficult to read the manuscript. Please correct these errors.

                        The manuscript underwent extensive editing of the English language and style by a native English speaker.